# ADVERSARIAL MASKED AUTOENCODER PURIFIER WITH DEFENSE TRANSFERABILITY

## ABSTRACT

The study of adversarial defense still struggles to combat with advanced adversarial attacks. In contrast to most prior studies that rely on the diffusion model for test-time defense to remarkably increase the inference time, we propose Masked AutoEncoder Purifier (MAEP), which integrates Masked AutoEncoder (MAE) into an adversarial purifier framework for test-time purification. While MAEP achieves promising adversarial robustness, it particularly features model defense transferability and attack generalization without relying on using additional data that is different from the training dataset. To our knowledge, MAEP is the first study of adversarial purifier based on MAE. Extensive experimental results demonstrate that our method can not only maintain clear accuracy with only a slight drop but also exhibit a close gap between the clean and robust accuracy. Notably, MAEP trained on CIFAR10 achieves state-of-the-art performance even when tested directly on ImageNet, outperforming existing diffusion-based models trained specifically on ImageNet.

## 1 INTRODUCTION

Proliferation of deep learning models across various domains has raised a pressing concern: Vulner-ability of these models to adversarial attacks (Athalye et al., 2018; Croce & Hein, 2020a; Kurakin et al., 2018; Madry et al., 2018) that aim to make the model behave abnormally by manipulating the input data with imperceptible perturbations.

In response to these threats, researchers have actively investigated techniques to enhance the robust-ness of machine learning models against adversarial attacks in two branches: (1) One promising paradigm involves the integration of adversarial training (Gowal et al., 2021; Hsiung et al., 2023; Huang et al., 2023a; Shafahi et al., 2019; Wang et al., 2023; You et al., 2023; Rebuffi et al., 2021a; Wang et al., 2019; Wu et al., 2020; Suzuki et al., 2023) during model training, where both the clean and adversarially perturbed data are used for training to improve robustness. While numerous studies have explored adversarial training, a notable disparity (from RobustBench) persists between the natural accuracy and robust accuracy. (2) Another paradigm is adversarial purification (Alfarra et al., 2022; Ho & Vasconcelos, 2022), which aims to detect and remove adversarial perturbations from input data before being fed into the model. The benefit of adversarial purification is that the downstream task like classifier is not needed to be retrained and that can generalize to different attacks at test time. The widely recognized adversarial purifiers (Nie et al., 2022; Wang et al., 2022; Wu et al., 2022; Zhang et al., 2023) are based on diffusion models. They introduce noises to the input images in the forward process and remove both the noises and adversarial perturbations in the reverse process. Although diffusion models are generalized to different attacks, called "attack generalization," they should learn the data distribution at first and pay the price of losing "defense transferability" (*i.e.*, transferability to other datasets). In addition, they should tune the hyper-parameters, including diffusion timestep (Nie et al., 2022) and guidance scale (Wang et al., 2022), carefully in different datasets and tasks.

In contrast to most prior studies that rely on the diffusion model, we propose Masked AutoEncoder Purifier (MAEP), which integrates Masked AutoEncoder (MAE) (He et al., 2022) into an adversarial purifier framework for test-time purification, as illustrated in Fig. 1, and elucidate the feasibility of this paradigm in Section 4.1. Specifically, in recent researches, MAE has leveraged the principles of self-supervised learning and masking image encoding to learn patch representations. While MAE

| Method | Defense type | Model | Additional data | Defense transferability | Attack generalization |
|---|---|---|---|---|---|
| TRADES (Zhang et al., 2019) | Adversarial training | Classifier | x | x | v |
| DiffPure (Nie et al., 2022) | Adversarial purifier | Diffusion model | x | x | v |
| ScoreOpt (Zhang et al., 2023) | Adversarial purifier | Diffusion model | x | x | v |
| DISCO (Ho & Vasconcelos, 2022) | Adversarial purifier | LIIF (Chen & Zhang, 2019) (EDSR (Lim et al., 2017) + local implicit model) | v | v | v |
| Anti-Adv (Alfarra et al., 2022) | Adversarial purifier | Classifier | x | v | v |
| NIM-MAE (You et al., 2023) | Adversarial training | Noise image modeling + ViT (Dosovitskiy et al., 2020) | x | x | v |
| Huang et al. (2023b) | Adversarial purifier | ViT-based model (Bao et al., 2021; Dong et al., 2023; He et al., 2022) | x | x | - |
| DRAM (Tsai et al., 2023) | Detection + Purifier | Masked AutoEncoder (He et al., 2022) | x | v | v |
| MAEP (Ours) | Adversarial purifier | Masked AutoEncoder (He et al., 2022) | x | v | v |

Table 1: Comparison among adversarial defenses. Only MAEP and Anti-Adv are "adversarial purifier" and possess defense transferability and attack generalization without needing additional data.

has demonstrated impressive performance across various vision tasks, its application in addressing adversarial defense problems remains relatively unexplored.

Table 1 summarizes the characteristics of MAEP and related studies. Our MAEP is the first study on exploring masked autoencoder for adversarial purifier. Specifically, MAEP is different from Huang *et al.* (Huang et al., 2023b) in that the latter only studied the robustness of a classifier with the structure of a ViT-based model but we investigate how to integrate the masking mechanism into an adversarial purifier instead of a classifier. Moreover, DRAM (Tsai et al., 2023) uses an MAE encoder as a test-time detection model and then repairs the image by the similar concept of Anti-Adv (Alfarra et al., 2022) if MAE detects that the input is an adversarial sample. This joint procedure of detection and repair process is different from our pure purifier framework. We also note that DRAM exhibits a lower robust accuracy and its defense capability is far below the SOTA adversarial defense methods. As for DISCO (Ho & Vasconcelos, 2022), it used EDSR (enhanced deep super-resolution network) (Lim et al., 2017) trained on additional dataset to extract the image latents and purify the adversarial image by the local implicit model in the latent space, so that its performance will heavily depend on additional data. As for NIM-MAE (You et al., 2023), it uses noise injection instead of a masking mechanism and can be viewed as an adversarial training approach, which is different from our MAEP.

The main contributions of this paper include:

- Unlike diffusion model-based adversarial purifiers, we are the first to explore integrating both the masking mechanism and purification as a novel defense paradigm. The structure of our MAEP is based on the Vision Transformer (ViT), paving the way for future work that applies the NLP and ViT-based concepts to enhance the defense capability.

- Our MAEP, an MAE-based adversarial purifier, offers both defense transferability and attack generalization without requiring additional data for training. We demonstrate that MAEP effectively transfers from a low-resolution dataset like CIFAR10 to a high-resolution dataset like ImageNet, while achieving defense robustness better than SOTA methods.

## 2 RELATED WORKS

### 2.1 ADVERSARIAL PURIFICATION

Anti-Adv (Alfarra et al., 2022) introduces an anti-adversary layer designed to steer the image $x$ away from the decision boundary. The perturbation direction is guided by the image prediction via classifier, given the absence of true labels during inference. However, it depends on if an adversarial image can be classified correctly, which is still challenging, potentially leading to misdirection.

DISCO (Ho & Vasconcelos, 2022) uses the concept of LIIF (Chen & Zhang, 2019) to extract the per-pixel feature by a pre-trained EDSR (Lim et al., 2017), and its training loss only needs to consider purification loss. DISCO is able to achieve acceptable robust accuracy and model transferability across different datasets.

DiffPure (Nie et al., 2022) employs a diffusion model for image purification and provides a theoretical guarantee: By introducing sufficient Gaussian noises in the forward process, adversarial perturbations can be effectively eliminated. Regardless of the classifiers or attacks, DiffPure remains effective with

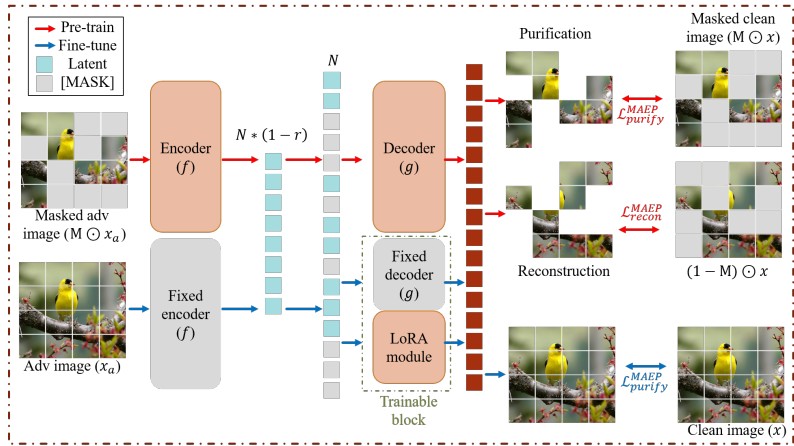

Figure 1: Workflow of our method. (a) Pre-training stage: Learn the patch representation by masking patch prediction and reconstruction by the purification loss. (b) Finetuning stage: Alleviate the information loss caused by masked patches in the pre-training stage (a.k.a the train-test discrepancy).

the caveat that the diffusion timestep must strike a balance. Actually, it should be large enough to remove adversarial perturbations yet small enough to preserve global label semantics.

Following DiffPure, ScoreOpt (Zhang et al., 2023) introduces the idea of score-based priors into diffusion-based adversarial purifier. Adversarial samples will converge towards points with the local maximum likelihood of posterior distribution, which is defined by pre-trained score-based priors.

## 2.2 MASKED AUTOENCODER (MAE)

MAE (He et al., 2022) implements a masking mechanism to enhance the performance of ViT (Dosovitskiy et al., 2020). Inspired by the Masked Language Modeling (MLM) technique used in BERT (Kenton & Toutanova, 2019), MAE operates as a pre-training model, focusing on learning patch representations during the pre-training stage and fine-tuning for downstream tasks. The pre-training objective involves reconstructing images by masking partial patches, facilitating the learning of meaningful patch representations. Hereafter, MAE and MLM will be interchangeably used.

Recently, some works (Huang et al., 2023b; Tsai et al., 2023; You et al., 2023) have employed MAE for the problems of demanding robustness. Huang *et al.* (Huang et al., 2023b) studied the robustness of ViT-based models (Dosovitskiy et al., 2020), including PeCo (Dong et al., 2023), BEiT (Bao et al., 2021), and MAE. However, the authors only focused on the adversarial perturbation with only small settings of attack budget, which is not enough in the challenging problem of the adversarial robustness. DRAM (Tsai et al., 2023) proposes a test-time detection method to repair adversarial samples in that the MAE reconstruction loss is directly used to detect the adversarial samples due to the assumption of different distributions of clean and adversarial samples. DRAM achieves robustness against adaptive attacks with the limitation that the adversarial sample should be close to the original sample with the mean-square error that is far smaller than those in existing works so as to weaken the adaptive attacks. NIM-MAE (You et al., 2023) uses the MAE structure to achieve adversarial training by injecting the noise into the entire image instead of masking patches within an image.

## 3 PRELIMINARY

### 3.1 NOTATION

Frequently used notations are defined as follows: Clean image $x$ and its corresponding label $y$; adversarial image $x_a$; classifier $c$, which outputs the logit; an image $x \in \mathbb{R}^{H \times W}$ cropped into $N$ patches of area $ps \times ps$ (w.r.t. patch size $ps$) before forwarding to purifier; purifier $\mathcal{P} = g \circ f$ with MAE encoder $f$ and MAE decoder $g$; masking ratio $r$ of MAE and corresponding binary mask $M$; and $\mathbf{1}$ is a matrix with all elements of 1.

## 3.2 ADVERSARIAL ATTACK

Given a classifier $c$ parameterized by $\theta$, and a clean data pair $(x, y)$, an adversarial attack aims to find an adversarial sample $x_a$ derived from $x$ to deceive the classifier (*i.e.*, $c(x_a) \neq y$) by the optimization as: $\max_{x_a} L(x_a, y; \theta), s.t. ||x_a - x||_p < \epsilon$, where $L$ is the training loss function, $|| \cdot ||_p$ denotes $p$-norm, and $\epsilon$ means the attack budget.

## 3.3 MASKED AUTOENCODER

Given an image $x \in \mathbb{R}^{H \times W}$ and a binary mask $M \in \mathbb{R}^{H \times W}$, the goal of MAE (He et al., 2022) is to reconstruct the entire image $\tilde{x} = x \in \mathbb{R}^{H \times W}$ from partial image $M \odot x$ with the reconstruction loss defined as:

$$L_{MAE}(x, \tilde{x}) = ||(\mathbf{1} - M) \odot \tilde{x} - (\mathbf{1} - M) \odot g \circ f(M \odot x)||_2, \tag{1}$$

where $\mathbf{1}$ is a matrix equal to $1^{H \times W}$, $\odot$ is the element-wise product of two matrices of the same size, $f$ is MAE encoder, $g$ is MAE decoder, and $M$ is a random binary image mask parameterized by the masking ratio $r$ as $||M||_1 = (1 - r) \times (H \times W)$. The masking ratio $r = 0$ if there is no masking.

## 4 PROPOSED METHOD

Masked Language Modeling (MLM) (Kenton & Toutanova, 2019) has demonstrated its effectiveness across numerous vision and language tasks. The fundamental concept involves masking certain words within sentences or patches within images, prompting the model to reconstruct the masked portions based on the unmasked context. This process facilitates the learning of meaningful relationships between words/patches and their representations. While MLM has consistently demonstrated state-of-the-art performance in recent studies, its application in adversarial purifiers remains underexplored. In this paper, inspired by our findings in Section 4.1 and Section 4.2, we propose a new adversarial defense paradigm of integrating MLM into a purifier in Section 4.3 and demonstrate its superiority over directly employing MLM in Section 4.4. Here, we will describe and verify the steps in MAEP.

### 4.1 MOTIVATION: DEFENSE TRANSFERABILITY VIA EMPIRICAL OBSERVATIONS

The predominant purification-based approaches (Nie et al., 2022; Wang et al., 2022; Wu et al., 2022; Zhang et al., 2023) in adversarial defense involve the diffusion model, which primarily modify the reverse diffusion process of DiffPure (Nie et al., 2022) using the same pre-trained model. However, these approaches exhibit limited transferability of the purifier, as depicted in Tables 2 and 3. The empirical evidence in Table 2 indicates that while DiffPure maintains a close gap between the clean accuracy and robust accuracy on CIFAR10, it experiences a significant drop (from $89.45\%$ to $69.00\%$) in robust accuracy when transferred to CIFAR100. A similar phenomenon can also be observed in Table 3. We exemplify DiffPure as the representative of diffusion-based adversarial defenses because other methods share a similar concept by primarily modifying the reverse diffusion process.

Most importantly, in real-world applications, it is impractical to always have access to a well-trained diffusion model, and training one from scratch is inefficient, especially when dealing with small datasets or when substantial resources are required for multiple datasets. Moreover, diffusion model-based defenses Nie et al. (2022)Zhang et al. (2023) suffer a significant drop in robust accuracy even when the training and testing datasets are closely related like CIFAR10 and CIFAR100, indicating a lack of defense generalization and vulnerability to images with minor differences.

The above observations motivate us to propose a new method that can achieve state-of-the-art robust accuracy and satisfy model transferability at the same time. Note that, even for DISCO that has been recognized to possess defense transferability, the clean accuracy of our MAEP is obviously higher than that of DISCO no matter transferability is considered or not. Please see Sec. 5.3 for more results.

### 4.2 PURIFICATION LOSS VS. CLEAN ACCURACY

In the literature, DISCO (Ho & Vasconcelos, 2022) is found to achieve both acceptable clean and robust accuracy by employing just one purification loss, while preserving model transferability. The

| Model | Training Data | | Test Data | | Clean Acc. (%) | Robust Acc. (%) | Avg. Acc. (%) |
|---|---|---|---|---|---|---|---|
| | CIFAR10 | CIFAR100 | CIFAR10 | CIFAR100 | | | |
| WRN28-10 | v | | v | | 94.78 | 0 | 47.39 |
| + DiffPure (Nie et al., 2022) | v | | v | | 89.58 | 89.45 | 89.51 |
| + DISCO (Ho & Vasconcelos, 2022) | v | | v | | 89.26 | 85.33 | 87.29 |
| + MAEP | v | | v | | 92.30 | 88.73 | **90.51** |
| + DiffPure (Nie et al., 2022) | | v | v | | 94.50 | 69.00 | 81.75 |
| + DISCO (Ho & Vasconcelos, 2022) | | v | v | | 89.78 | 87.44 | **88.61** |
| + MAEP | | v | v | | 91.58 | 84.73 | 88.16 |

Table 2: Transferability of adversarial defenses (DISCO, Diffpure, and our MAEP) from CIFAR100 (Krizhevsky et al., 2009a) to CIFAR10 (Krizhevsky et al., 2009a) (trained on CIFAR100 but tested on CIFAR10) under WRN28-10 (Zagoruyko & Komodakis, 2017) and AutoAttack (Croce & Hein, 2020a) with attack budget $\epsilon_\infty = 8/255$. Avg. Acc. is the average of clean acc. and robust acc. and used to show overall performance.

| Model | Training Data | | Test Data | | Clean Acc. (%) | Robust Acc. (%) | Avg. Acc. (%) |
|---|---|---|---|---|---|---|---|
| | CIFAR10 | CIFAR100 | CIFAR10 | CIFAR100 | | | |
| WRN28-10 | | v | | v | 81.66 | 0 | 40.83 |
| + DiffPure (Nie et al., 2022) | | v | | v | 61.98 | 61.19 | 61.58 |
| + DISCO (Ho & Vasconcelos, 2022) | | v | | v | 69.78 | 76.91 | 73.34 |
| + MAEP | | v | | v | 73.67 | 76.22 | **74.95** |
| + ScoreOpt-O (Zhang et al., 2023) | v | | | v | 57.55 | 42.83 | 50.19 |
| + ScoreOpt-N (Zhang et al., 2023) | v | | | v | 54.87 | 54.37 | 54.62 |
| + DiffPure (Nie et al., 2022) | v | | | v | 81.00 | 40.00 | 60.50 |
| + DISCO (Ho & Vasconcelos, 2022) | v | | | v | 72.50 | 69.22 | 70.86 |
| + MAEP | v | | | v | 75.37 | 68.75 | **72.06** |

Table 3: Transferability of adversarial defenses (ScoreOpt, Diffpure, DISCO, and our MAEP) from CIFAR10 to CIFAR100 (trained on CIFAR10 but tested on CIFAR100) under WRN28-10 and AutoAttack with attack budget $\epsilon_\infty = 8/255$.

purification loss of DISCO is defined as:

$$L_{purify}^{DISCO}(x, \mathcal{P}(x_a)) = \ell_1(x, \mathcal{P}(x_a)), \tag{2}$$

where $\mathcal{P}$ is a purifier used to purify input image $x_a$ by reconstructing the clean image $x$ in terms of $\ell_1$-norm loss between $x$ and $x_a$.

Specifically, DISCO shows that it can purify the adversarial image efficiently, expressed as:

$$\mathcal{P}(x_a) \approx x, \tag{3}$$

$$c(\mathcal{P}(x_a)) \approx c(x), \tag{4}$$

where Eq. (3) denotes the perceptual similarity between the clean image $x$ and purified image $\mathcal{P}(x_a)$, Eq. (4) indicates label-preservation, and $c$ is a pre-trained classifier from RobustBench or PyTorch official website. Nevertheless, we can observe from Tables 2 and 3 that, compared with MAEP, there is still a room for DISCO to improve label-preservation for purified clean images, defined as:

$$c(\mathcal{P}(x)) \approx c(x). \tag{5}$$

Although DISCO (Ho & Vasconcelos, 2022) does not ensure to preserve the clean accuracy, it indeed shows good trade-off between the clean accuracy and robust accuracy in several testing scenarios, including different classifiers, different attack algorithms (such as Autoattack (Croce & Hein, 2020a), PGD (Madry et al., 2018), FAB (Croce & Hein, 2020b), BIM (Kurakin et al., 2018), BPDA (Athalye et al., 2018), and FGSM (Goodfellow et al., 2015)), and transferability to different datasets.

Based on the above observations, we conjecture that the purification loss ($\ell_1(\mathcal{P}(x_a), x)$ in Eq. (2)) can efficiently purify the adversarial image without remarkably sacrificing clean accuracy. Here, we provide a feasible but simple explanation of why DISCO can have acceptable performance without needing to consider $\ell_1(\mathcal{P}(x), x)$, as illustrated in Fig. 2.

Specifically, based on the assumption that an adversarial image is created to be similar to its clean counterpart in term of $L_\infty$-norm as:

$$x \approx x_a \ s.t. \ |x - x_a|_\infty < \epsilon, \tag{6}$$

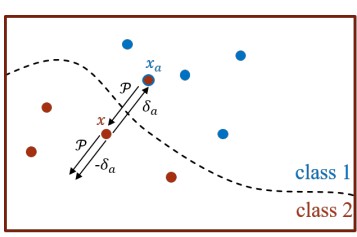

Figure 2: Purification loss learns the direction from $x_a$ to $x$ and the direction of $-\delta_a$ (one-step adversarial perturbation along negative gradient) is roughly the same as $\mathcal{P}(x_a)$ due to Eq. (6).

| | Acc. (%) | Formula |
|---|---|---|
| $c(\mathcal{P}(x_a))$ | 95.16 | Eq. (4) |
| $c(\mathcal{P}(x))$ | 89.94 | Eq. (8) |

Table 4: Verification of our conjecture on the purification loss ($\ell_1(\mathcal{P}(x_a), x)$ in Eq. (2). For $c(\mathcal{P}(x)) \approx c(x - \delta_a)$, we pre-process the training dataset by adding $-\delta_a$ (via PGD) and feeding it to a non-defense classifier (ResNet-18) pre-trained on CIFAR-10 for testing. Since our derivation introduces approximation to remove the influence of purifier and is only based on a classifier, it means that this is a theoretical accuracy of a purifier and Eq. (2) can properly maintain clean accuracy.

we can derive

$$\mathcal{P}(x) - x \approx \mathcal{P}(x_a) - x_a = -\delta_a, \tag{7}$$

where the direction of purification, $\mathcal{P}(x) - x$, for clean image $x$ is similar to $\mathcal{P}(x_a) - x_a$ of an adversarial image due to the adversarial condition specified in Eq. (6).

In practice, input images fall into two categories: adversarial images ($x_a$) and clean images ($x$). The robust accuracy of purifier is tied to $x_a$ and the prediction of $x_a$ can be calculated by $c(\mathcal{P}(x_a)) \approx c(x)$, as detailed in Eq. (4). The clean accuracy of purifier, which DISCO didn't make a discussion, is related to $x$ and the prediction of $\mathcal{P}(x)$ can be derived from Eq. (7) as:

$$c(\mathcal{P}(x)) = c(x + (\mathcal{P}(x) - x)) \approx c(x + (\mathcal{P}(x_a) - x_a)) = c(x - \delta_a), \tag{8}$$

where $\mathcal{P}(x_a) - x_a$ is roughly equal to $-\delta_a$, as illustrated in Fig. 2. To verify the above conjecture, Table 4 shows the results. Thus, the goal of boosting defense transferability and maintaining clean accuracy can be better realized by combining the purification loss (Eq. (2)) and MLM together.

### 4.3  OBJECTIVE FUNCTION DESIGN

We study diverse objective function designs (He et al., 2022; Ho & Vasconcelos, 2022; Kenton & Toutanova, 2019; Shafahi et al., 2019; Zhang et al., 2019), all geared towards enhancing robust accuracy. These designs include TRADES (Zhang et al., 2019), MLM (He et al., 2022; Kenton & Toutanova, 2019), and reconstruction loss (Ho & Vasconcelos, 2022; Shafahi et al., 2019), where TRADES is a popular choice for training robust classifiers, MLM showcases effectiveness in vision tasks, and reconstruction loss is often used for image reconstruction. We will delve into a discussion of the results produced by these designs, emphasizing the superior performance of our MAEP technique among these alternatives. In the following discussions, the distance measure $D$ is chosen to be $\ell_1$-norm. Please refer to Sec. 7 of Appendix for details. The descriptions regarding reconstruction loss and extensions based on TRADES will be shown in Sec. 8 of Appendix, and the discussions on MAE and MAEP will be described here. In addition, the performance comparison of all loss designs can be found in Table 12 of Sec. 8 in Appendix.

#### 4.3.1  MASKED LANGUAGE MODELING (MLM)

MLM (He et al., 2022) has recently garnered considerable success in various tasks, such as BERT (Kenton & Toutanova, 2019) and MAE (He et al., 2022). We leverage this concept directly to train the purifier through a two-step process. Initially, it aims to learn adversarial embeddings during the pre-training stage, as indicated in Eq. (1), and subsequently finetunes to purify an adversarial image. The loss function is described as follows with respect to the two-step process:

**a) Pre-training stage**

$$L_{pre-train} = L_{MAE}(x_a, x) = \|(\mathbf{1} - M) \odot x - (\mathbf{1} - M) \odot (g \circ f(M \odot x_a))\|. \tag{9}$$

**b) Finetuning stage**

$$L_{finetune} = D(\mathcal{P}(x_a), x). \tag{10}$$

### 4.3.2 TOTAL LOSS IN MAEP

Unlike the aforementioned designs, our method, MAEP, is investigated to integrate both the MLM and purification loss of DISCO in Eq. (2) to boost adversarial robustness and model transferability. We also show that solely utilizing MAE/MLM does not yield satisfactory performance.

The entire loss of MAEP can be separated into two parts. Part 1) The purify loss, adapted from Eq. (2), addresses the purification task focusing solely on the unmasked region within an image. Several reasons support this decision to exclusively handle the unmasked region. Firstly, training with partial image information can generalize to the entire image via position embedding, as outlined in MAE (He et al., 2022). In addition, the purifier is designed to operate on the entire image rather than the masked region. Secondly, the incorporation of MLM discussed in Part 2) below can effectively address the issue of dealing with the masked region. Lastly, a finetuning strategy, as will be discussed in Section 4.4, is introduced to refine the gap between the results obtained from the unmasked and masked regions. Part 2) Compared to those using the unmasked region in Part 1) for purification, the MLM reconstructs the masked region of an image by the unmasked region. This masking mechanism will help the model learn the adversarial representation or recognize the adversarial perturbation so as to further boost the performance.

Based on the above concerns, it is ready to define the MAEP loss. First, we define the purification loss. Recall that $g \circ f$ is called the purifier in MAE. To ensure the clean accuracy and robust accuracy, as described in Section 4.2, we adopt Eq. (2) for masked region and the purification loss of unmasked region in MAEP is defined as:

$$L_{purify}^{MAEP} = \|M \odot x - M \odot g \circ f(M \odot x_a)\|. \tag{11}$$

Note that $L_{purify}^{MAEP}$ reconstructs the clean image $x$ based on the adversarial image $x_a$, which is consistent with Part 1) and is different from the traditional MAE, as shown in Eq. (1).

Second, following the reconstruction loss (Eq. (9)) in MAE, the reconstruction loss $L_{recon}^{MAEP}$ of the masked region in MAEP is defined as $L_{recon}^{MAEP} = L_{pre-train}$.

Therefore, the entire loss function of MAEP is derived as:

$$\begin{aligned} L^{MAEP} &= L_{purify}^{MAEP} + L_{recon}^{MAEP} \\ &= \|M \odot x - M \odot g \circ f(M \odot x_a)\| + \|(\mathbf{1} - M) \odot x - (\mathbf{1} - M) \odot g \circ f(M \odot x_a)\| \\ &\geq \|x - g \circ f(M \odot x_a)\|, \end{aligned} \tag{12}$$

where the equal sign holds when $\|\cdot\|$ is $\ell_1$-norm, which is adopted as the distance measure in MAEP. The masking ratio $r \in (0, 1)$ controls the image mask $M$. Eq. (12) will degenerate to the purification loss in Eq. (2) when $r = 0$.

### 4.4 TRAIN-TEST DISCREPANCY AND TRANSFER LEARNING

Recall that representation learning, such as BERT and MAE, often adopts two-stage training, which learns the meaningful latent representation in the pre-training stage and boosts the performance of downstream tasks in the finetuning stage, as shown in Eqs. (9) and (10), respectively. So far, during pre-training, MAEP has purified the adversarial image (Eq. (12)), eliminating the need of finetuning. We will further consider an issue of train-test discrepancy within the masking mechanism and the practical concern of transferring to different datasets, so that we can use finetuning in MAEP as well.

Inspired by (Touvron et al., 2019), if the distribution of input images at inference time and that at training time are similar, the problem, called "train-test discrepancy," due to the different conditions at the training and testing stages, can be properly alleviated to improve performance. Additionally, the BERT-based methods (Dosovitskiy et al., 2020; He et al., 2022; Kenton & Toutanova, 2019) have good transferability by model finetuning. Actually, in MAEP, the masking ratio $r = 0.5$ in training time is different from $r = 0$ in inference time that will cause train-test discrepancy. Although this problem has been tackled by position embedding, as described in MAE (He et al., 2022), we find that finetuning MAEP to alleviate the train-test discrepancy can further improve the performance. Based on the above concerns, we design a simple finetuning strategy to solve the train-test discrepancy and maintain the goodness of transferability.

The finetuning strategy here is simple in that we train an MAEP with masked images and then use LoRA (Hu et al., 2021) to only finetune the decoder with masking ratio $r = 0$. In practice, the pre-training ViT-based model (Dosovitskiy et al., 2020) and diffusion-based model (Ho et al., 2020) often require significant computing resources. In most usage in real life, we don't want to spend too much computing power to finetune a downstream task. That is why we adopt a lightweight finetuning technique (*i.e.,* LoRA) to finetune only a minimal number of trainable parameters. Here, the lightweight finetuning technique can be replaced by any other similar approaches, such as Ladder Side-Tuning(LST) (Sung et al., 2022), P-tuning(Liu et al., 2023), and Prefix-Tuning (Li & Liang, 2021). We can observe from Table 5 that the both the clean and robust accuracy can be increased via the use of finetuning. In addition, we further provide a straightforward comparison between the traditional finetuning methods and LoRA in Table 13 of Sec. 9 of Appendix. The results indicate that our MAEP can achieve both the dataset transfer and train-test discrepancy alleviation.

| Model | Pre-train | | Finetune | | Clean Acc. (%) | Robust Acc. (%) | Avg. Acc. |
|-------|-----------|---|----------|---|----------------|-----------------|-----------|
| | CIFAR10 | CIFAR100 | CIFAR10 | CIFAR100 | | | |
| WRN28-10 | v | - | - | - | 94.78 | 0 | 47.39 |
| | v | - | - | - | 92.30 | 88.73 | 90.52 |
| | v | - | v | - | 92.13 | 89.40 | 90.77 |
| + MAEP | v | v | - | - | 91.58 | 84.73 | 88.16 |
| | - | v | v | - | 91.74 | 85.55 | 88.65 |
| WRN28-10 | - | v | - | - | 81.66 | 0 | 40.83 |
| | - | v | - | - | 73.67 | 76.22 | 74.95 |
| | - | v | - | v | 73.57 | 76.47 | 75.02 |
| + MAEP | v | - | - | - | 75.37 | 68.75 | 72.06 |
| | v | - | - | v | 75.81 | 70.37 | 73.09 |

Table 5: Performance of finetuning and transferability of MAEP under AutoAttack with $\epsilon_\infty = 8/255$.

## 5 EXPERIMENTS

In the previous sections, we have verified the proposed steps in MAEP. In this section, we will provide entire evaluation and comparison with SOTA adversarial defenses.

### 5.1 DATASETS, MODEL SETTINGS, AND IMPLEMENTATION DETAIL

Three commonly used datasets, CIFAR10 (Krizhevsky et al., 2009b), CIFAR100 (Krizhevsky et al., 2009b), and ImageNet (Deng et al., 2009), were adopted. All models were trained using NVIDIA V100 GPUs. Details of model structure and parameter settings can be found in Sec. 10 of Appendix.

For model architectures, we employed WRN-28-10 (Zagoruyko & Komodakis, 2016) and its corresponding model weights provided by RobustBench for CIFAR10. However, for CIFAR100 and ImageNet, due to the absence of model weights from RobustBench, we adopted them from DISCO (Ho & Vasconcelos, 2022) and PyTorch, respectively. For the attacks, we consider PGD-$\ell_\infty$ and AutoAttack, and set the permissible perturbation $\epsilon$ such that $|\epsilon|_\infty \leq 8/255$.

For training the purifier, we pre-trained the MAEP from scratch with the loss in Eq. (12) and set the masking ratio $r = 0.5$. To finetune MAEP, we adopted LoRA (Hu et al., 2021). During inference, the encoder of MAEP remains fixed while the decoder is also fixed but combined with a parallel trainable LoRA module, as illustrated in Fig. 1. The masking ratio, initially set to $r = 0.5$, was adjusted to $r = 0$ for downstream tasks, resulting in the loss calculation relying solely on purification loss $L_{purify}^{MAEP}$, as reconstruction loss $L_{recon}^{MAEP}$ becomes zero. During inference, each result for clean or robust accuracy was obtained by averaging from 5 runs with varying random seeds.

### 5.2 EVALUATION OF ADVERSARIAL DEFENSE

We adopted SOTA adversarial purifiers for comparison, including diffusion model-based approaches (Nie et al., 2022; Zhang et al., 2023; Yoon et al., 2021) and non-diffusion-based approaches (Alfarra et al., 2022; Ho & Vasconcelos, 2022; Shi et al., 2021; Hill et al., 2020; Wu et al., 2020). For DiffPure (Nie et al., 2022), we used the official code and tested the purifier under the same experimental setup mentioned in Section 5.1. For ScoreOpt (Zhang et al., 2023), the classifier was trained by the authors and not from RobustBench. For a fair comparison, we used the official code and only replaced the

default classifier of ScoreOpt with the WRN-28-10 model provided by RobustBench. DRAM (Tsai et al., 2023) was not selected for comparison because DRAM involved the detection of adversarial samples that is not required in our framework.

Table 6 shows the comparison results with dataset CIFAR-10. We have observations as follows: (1) For robust accuracy, our MAEP and ScoreOpt-O (Zhang et al., 2023) are comparable but better than DiffPure (Nie et al., 2022). However, for clean accuracy, MAEP performs better than (Nie et al., 2022)(Zhang et al., 2023). (2) MAEP and diffusion-based defenses are generally better than other methods in terms of average accuracy. For AutoAttack with budget of $\ell_2$-norm, please see Table 18.

| Defense Methods | Clean Accuracy (%) | Robust Accuracy (%) | Average Accuracy (%) | Attacks |
|---|---|---|---|---|
| No defense | 94.78 | 0 | 47.39 | PGD-$\ell_\infty$/AutoAttack (Standard) |
| AWP (Wu et al., 2020)* | 88.25 | 60.05 | 74.15 | AutoAttack (Standard) |
| Anti-Adv (Alfarra et al., 2022)* + AWP | 88.25 | 79.21 | 83.73 | AutoAttack (Standard) |
| DISCO (Ho & Vasconcelos, 2022) | 89.26 | 82.99 | 86.12 | PGD-$\ell_\infty$ |
| DISCO (Ho & Vasconcelos, 2022) | 89.26 | 85.33 | 87.29 | AutoAttack (Standard) |
| DiffPure (Nie et al., 2022) | 88.62±2.12 | 87.12±2.45 | 87.87±2.28 | PGD-$\ell_\infty$ |
| DiffPure (Nie et al., 2022) | 88.15±2.70 | 87.29±2.45 | 87.72±2.57 | AutoAttack (Standard) |
| ScoreOpt-N (Zhang et al., 2023) | 91.03 | 80.04 | 85.53 | PGD-$\ell_\infty$ |
| ScoreOpt-O (Zhang et al., 2023) | 89.16 | 89.15 | 89.15 | PGD-$\ell_\infty$ |
| ScoreOpt-N (Zhang et al., 2023) | 91.31 | 81.79 | 86.55 | AutoAttack (Standard) |
| ScoreOpt-O (Zhang et al., 2023) | 89.18 | 89.01 | 89.09 | AutoAttack (Standard) |
| SOAP* (Shi et al., 2021) | 96.93 | 63.10 | 80.01 | PGD-$\ell_\infty$ |
| Hill *et al.* (Hill et al., 2020)* | 84.12 | 78.91 | 81.51 | PGD-$\ell_\infty$ |
| ADP ($\sigma = 0.1$) (Yoon et al., 2021)* | 93.09 | 85.45 | **89.27** | PGD-$\ell_\infty$ |
| MAEP | 92.31 | 86.19 | 89.25 | PGD-$\ell_\infty$ |
| MAEP (W/ LoRA) | 92.13 | 87.09 | **89.61** | PGD-$\ell_\infty$ |
| MAEP | 92.30 | 88.73 | **90.52** | AutoAttack (Standard) |
| MAEP (W/ LoRA) | 92.13 | 89.40 | **90.77** | AutoAttack (Standard) |

Table 6: Robustness evaluation and comparison. Classifier: WRN-28-10. Testing dataset: CIFAR-10. Asterisk (*) indicates that the results were excerpted from the papers.

For CIFAR100, the robustness comparison results are shown in Table 7. Different from CIFAR-10, under CIFAR-100, DISCO performs better than DiffPure for both clean and robust accuracy, and MAEP outperforms DISCO and DiffPure with a large gap. It is noteworthy that for both MAEP and DISCO, their robust accuracy is higher than clean accuracy. One possible explanation is that they primarily learn the mapping from the adversarial image $x_{adv}$ to the clean image $x$. This conforms to the verification in Table 4 and depicts that the theoretical optimal situation, in which the robust accuracy is higher than clean accuracy, may exist.

| Defense Methods | Clean Accuracy (%) | Robust Accuracy (%) | Average Accuracy (%) | Attacks |
|---|---|---|---|---|
| No defense | 81.66 | 0 | 40.83 | PGD-$\ell_\infty$/AutoAttack (Standard) |
| Rebuffi *et al.* (Rebuffi et al., 2021b) | 62.41 | 32.06 | 47.23 | AutoAttack (Standard) |
| Wang *et al.* (Wang et al., 2023) | 72.58 | 38.83 | 55.70 | AutoAttack (Standard) |
| Cui *et al.* (Cui et al., 2023) | 73.85 | 39.18 | 56.51 | AutoAttack (Standard) |
| DISCO (Ho & Vasconcelos, 2022) | 69.78 | 73.36 | **71.57** | PGD-$\ell_\infty$ |
| DISCO (Ho & Vasconcelos, 2022) | 69.78 | 76.91 | 73.34 | AutoAttack (Standard) |
| DiffPure (Nie et al., 2022) | 61.96±2.26 | 59.27±2.95 | 60.61±2.60 | PGD-$\ell_\infty$ |
| DiffPure (Nie et al., 2022) | 61.98±2.47 | 61.19±2.87 | 61.58±2.67 | AutoAttack (Standard) |
| MAEP | 73.67 | 70.96 | **71.57** | PGD-$\ell_\infty$ |
| MAEP (w/ LoRA) | 73.57 | 71.14 | **72.36** | PGD-$\ell_\infty$ |
| MAEP | 73.67 | 76.22 | **74.95** | AutoAttack (Standard) |
| MAEP (w/ LoRA) | 73.57 | 76.47 | **75.02** | AutoAttack (Standard) |

Table 7: Robustness evaluation and comparison.Classifier: WRN-28-10. Testing dataset: CIFAR-100.

## 5.3 TRANSFERABILITY TO HIGH-RESOLUTION DATASET

In addition to Tables 2 and 3, transferability from low-resolution dataset to high-resolution dataset is examined in Table 8. With an attack budget of $\epsilon_\infty = 4/255$, MAEP achieves approximately 74% accuracy when transferring from CIFAR10 to ImageNet, outperforming both DiffPure (68.60%) and ScoreOpt (68.05%). Notably, DiffPure and ScoreOpt were trained on ImageNet, while MAEP was trained solely on CIFAR10. When attack budget is increased to $\epsilon_\infty = 8/255$, our MAEP maintains promising accuracy, outperforms DiffPure and ScoreOpt, and remains comparable with DISCO at $\epsilon_\infty = 4/255$. This indicates again that our MAEP possesses powerful transferable defense capability.

Moreover, MAEP experiences only a slight 3% drop in clean accuracy with respect to the original classifier's accuracy 80.85% without defense, whereas diffusion-based approaches exhibit a decrease of around 10%. This difference attributes to the fact that diffusion models introduce noises to remove adversarial perturbations, thereby reducing clean accuracy.

| Model | Train | | Test | | Clean Acc. (%) | Robust Acc. (%) | Avg. Acc. (%) | Attacks |
|---|---|---|---|---|---|---|---|---|
| | CIFAR10 | ImageNet | CIFAR10 | ImageNet | | | | |
| ResNet50 | - | v | - | v | 80.85 | 0 | 40.42 | $\epsilon_\infty = 4/255$ |
| + MAEP | v | - | - | v | 77.84 | 70.62 | **74.23** | $\epsilon_\infty = 4/255$ |
| + MAEP (w/ LoRA) | v | - | - | v | 77.61 | 71.23 | **74.42** | $\epsilon_\infty = 4/255$ |
| + MAEP | v | - | - | v | 77.62 | 66.19 | **71.91** | $\epsilon_\infty = \mathbf{8/255}$ |
| + MAEP (w/ LoRA) | v | - | - | v | 77.61 | 67.44 | **72.53** | $\epsilon_\infty = \mathbf{8/255}$ |
| + DISCO (Ho & Vasconcelos, 2022) | v | - | - | v | 76.61 | 69.12 | 72.86 | $\epsilon_\infty = 4/255$ |
| + MAEP | - | v | - | v | 77.97 | 73.94 | **75.96** | $\epsilon_\infty = 4/255$ |
| + DISCO | - | v | - | v | 77.54 | 70.44 | 73.99 | $\epsilon_\infty = 4/255$ |
| + DiffPure* (Nie et al., 2022) | - | v | - | v | 70.01±12.18 | 67.11±12.03 | 68.60±12.10 | $\epsilon_\infty = 4/255$ |
| + ScoreOpt* (Zhang et al., 2023) | - | v | - | v | 70.07 | 66.02 | 68.05 | $\epsilon_\infty = 4/255$ |
| Res18 | - | v | - | v | | | | |
| + DISCO* (Ho & Vasconcelos, 2022) | - | v | - | v | 67.98 | 60.88±0.17 | 64.43 | $\epsilon_\infty = 4/255$ |
| WRN50-2 | - | v | - | v | | | | |
| + DISCO* (Ho & Vasconcelos, 2022) | - | v | - | v | 75.1 | 69.5±0.23 | 72.3 | $\epsilon_\infty = 4/255$ |

Table 8: Transferability of adversarial defenses (DISCO and our MAEP) from CIFAR10 to ImageNet (trained on CIFAR10 but tested on ImageNet) under ResNet50 (pre-trained model weight is from official PyTorch) and AutoAttack(Croce & Hein, 2020a). sterisk (*) indicates that the results were excerpted from the papers.

## 5.4 EVALUATION OF TRAINING AND INFERENCE TIME, AND PURIFICATION QUALITY

We compare the training and inference time for the adversarial defense methods, including DiffPure (Nie et al., 2022), ScoreOpt (Zhang et al., 2023), DISCO (Alfarra et al., 2022), and MAEP. The comparison results are shown in Tables 9 and 10. Although DiffPure did not specify the training time, DDPM (Ho et al., 2020) mentions a training duration of 10.6 hours on a TPU v3-8, indicating substantial computational cost. We can see that both MAEP and DISCO perform much faster than the other methods. We also examine the purification quality. Please see Sec. 11 of Appendix for details.

| Model | Inference Time Cost (sec) |
|---|---|
| MAEP | 0.00907 |
| MAEP (w/ LoRA) | 0.01012 |
| DISCO | 0.00480 |
| DiffPure ($t^*$=0.1) | 10.56000 |
| ScoreOpt-N (Steps=20) | 1.17000 |
| ScoreOpt-O (Steps=5) | 0.36000 |

Table 9: Inference time of MAEP and diffusion-based methods per image on CIFAR10. The steps employed by both ScoreOpt-N and ScoreOpt-O follow the recommendation provided by the authors.

| Model | Training Time Cost | GPUs |
|---|---|---|
| MAEP | ~1 day | Two V100 GPUs |
| MAEP (finetuned w/ LoRA) | ~10 minutes | Two V100 GPUs |
| DISCO | ~23 hours | Single V100 GPU |
| DiffPure ($t^*$=0.1) | 10.6 hours | TPU v3-8 (similar to 8 V100 GPUs) |
| ScoreOpt-N (Steps=20) | ~2 days | 8 V100 GPUs |
| ScoreOpt-O (Steps=5) | ~2 days | 8 V100 GPUs |

Table 10: Training time of MAEP and diffusion-based methods on CIFAR10.

## 6 CONCLUSION AND LIMITATION

Different from diffusion model-based adversarial defenses, we are first to explore how to integrate both the masking mechanism and purifier as a new defense paradigm. Our method is an MAE-based adversarial purifier and possesses the characteristics of defense transferability and attack generalization without needing additional data.

Compared to diffusion models, even without pre-trained model weights available for MAEP on well-known datasets, MAEP still delivers competitive performance across various datasets. Additionally, in the absence of pre-trained models, MAEP requires fewer training resources than diffusion-based approaches. Moreover, recent advancements in NLP, driven by Transformer-based models, have witnessed tremendous success. MAEP adopts the Vision Transformer (ViT) structure, indicating a promising direction for integrating large language models into adversarial purifiers.

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
