REPRODUCIBILITY

To ensure reproducibility, we will make the source code publicly available after acceptance.

APPENDIX

## 7 DISTANCE MEASUREMENT

In this section, we demonstrate why we use $\ell_1$-norm as our distance measurement. As shown in Table 11, we can see that $\ell_1$-norm outperforms the MSE. The main reason is that $\ell_1$-norm is more sensitive to the small difference, which is related to the adversarial condition described in Eq. (6), while MSE tends to punish the bigger errors. Thus, it is concluded that it is more feasible to adopt $\ell_1$-norm to eliminate the small perturbations caused by adversarial attacks.

| Distance Function | Clean Acc. (%) | Robust Acc. (%) | Attacks |
|---|---|---|---|
| $\ell_1$-norm | 92.30 | 88.73 | AutoAttack (Standard) |
| Mean Square Error (MSE) | 93.59 | 69.72 | AutoAttack (Standard) |

Table 11: Robustness evaluation and comparison between different distance functions. Classifier: WRN-28-10. Testing dataset: CIFAR-10.

## 8 MORE OBJECTIVE FUNCTION DESIGNS

**1) Reconstruction Loss.** Intuitively, to enhance both the clean accuracy and robust accuracy of an NN model, the purified/reconstructed image should closely resemble its clean version by minimizing the loss as:

$$L = D(\mathcal{P}(x_a), x) + D(\mathcal{P}(x), x), \tag{13}$$

where the first term denotes the distance between the purified adversarial image and its corresponding clean image, and the second term measures the distance between the purified clean image and true clean one.

**2) TRADES (Zhang et al., 2019).** TRADES proposed to train a robust classifier with the loss function defined as:

$$L = CrossEntropy(c(x), y) + KL(c(x), c(x_a))/\lambda, \tag{14}$$

where the first term maintains the clean accuracy while the second term focuses on improving the robust accuracy by making logits of adversarial sample similar to those of clean acccuracy, and $c$ is a classifier.

**2.1) TRADES in pixel domain.** To replicate the success of TRADES in adversarial training, we adapt its concept from training a robust classifier to training an adversarial purifier. The main difference is that the purifier needs to process the image instead of class prediction. Therefore, we replace the cross-entropy loss and KL divergence loss in Eq. (14) with a reconstruction loss in the image pixel domain to meet the purifier's requirement as:

$$L = D(\mathcal{P}(x), x) + D(\mathcal{P}(x), \mathcal{P}(x_a))/\lambda, \tag{15}$$

where the first term maintains the purified clean image quality and the second term tries to purify the adversarial image by mimicking the clean image in a sense similar to KL divergence loss in Eq. (14).

**2.2) TRADES in latent domain.** Unlike the methods proposed to concentrate on the image pixel domain, several works, such as Latent Diffusion (Rombach et al., 2022) and CLIP (Radford et al., 2021), have achieved notable success by processing image latent representations. In our approach, as indicated in Eq. (16) below, we maintain clean image purification (1st term) while enforcing constraints on adversarial perturbations within the latent space (2nd term) as:

$$L = D(\mathcal{P}(x), x) + D(f(x), f(x_a))/\lambda. \tag{16}$$

In Table 12, we provide a comparison of the performance of all loss designs discussed here to verify the design of MAEP. We have the following observations: (1) Although reconstruction loss concurrently learns the reconstruction of both clean and adversarial images, its performance falls short of DISCO, which concentrates solely on reconstructing adversarial images. (2) MLM and DISCO are closely associated with MAEP. Directly applying MLM appears ineffective, while MAEP demonstrates performance enhancement over DISCO. (3) Exploiting the concept of TRADES does not aid in learning an adversarial purifier. Thus, our validation shows that MAEP significantly outperforms other approaches. Note that, as discussed in Table 1, DISCO needs additoinal data but MAEP does not.

| Defenses | Clean Acc. (%) | Robust Acc. (%) | Avg. Acc. |
|---|---|---|---|
| WRN28-10 | 94.78 | 0 | 47.39 |
| + DISCO (Ho & Vasconcelos, 2022) | 89.26 | 85.33 | 87.29 |
| + Reconstruction (Eq. (13)) | 94.74 | 82.73 | 88.73 |
| + TRADES (pixel, Eq. (15)) | 94.75 | 0.85 | 47.81 |
| + TRADES (latent, Eq. (16)) | 94.64 | 38.16 | 66.40 |
| + MLM (He et al., 2022) | 92.85 | 61.46 | 77.15 |
| + MAEP | 92.30 | 88.73 | **90.52** |

Table 12: Performance comparison of different objective functions on CIFAR10 dataset under AutoAttack with attack budget $\epsilon_\infty = 8/255$.

## 9 MORE RESULTS ON FINETUNING

In this section, we provide more results on finetuning in Table 13 and Table 14.

| Defense Methods | Clean Acc. (%) | Robust Acc. (%) | Attacks |
|---|---|---|---|
| No defense | 94.78 | 0 | AutoAttack (Standard) |
| MAEP | 92.30 | 88.73 | AutoAttack (Standard) |
| MAEP (w/ LoRA) | 92.13 | 89.40 | AutoAttack (Standard) |
| MAEP (w/ finetune) | 92.51 | 84.70 | AutoAttack (Standard) |

Table 13: Robustness evaluation and comparison between LoRA and the traditional finetuning approach. Classifier: WRN-28-10. Testing dataset: CIFAR-10. Asterisk (*) indicates that the results were excerpted from the papers.

| Model | Pre-train | | Finetune | | Clean Acc. (%) | Robust Acc. (%) |
|---|---|---|---|---|---|---|
| | CIFAR10 | CIFAR100 | CIFAR10 | CIFAR100 | | |
| WRN28-10 | v | - | - | - | 94.78 | 0 |
| MAEP | v | - | - | - | 92.30 | 88.73 |
| MAEP (w/ LoRA) | v | - | v | - | 92.51 | 88.94 |
| MAEP (w/ finetune) | v | - | v | - | 92.51 | 84.70 |
| MAEP (w/ LoRA) | - | v | v | - | 91.57 | 84.55 |
| MAEP (w/ finetune) | - | v | v | - | 91.16 | 83.30 |
| WRN28-10 | - | v | - | - | 81.66 | 0 |
| MAEP | - | v | - | v | 73.67 | 76.22 |
| MAEP (w/ LoRA) | - | v | - | v | 73.56 | 76.25 |
| MAEP (w/ finetune) | - | v | - | v | 72.66 | 75.13 |
| MAEP (w/ LoRA) | v | - | - | v | 75.38 | 69.83 |
| MAEP (w/ finetune) | v | - | - | v | 73.05 | 74.50 |

Table 14: Performance of finetuning and transferability of MAEP on CIFAR10 and CIFAR100 datasets under AutoAttack with $\epsilon_\infty = 8/255$.

## 10  MODEL STRUCTURE AND PARAMETERS

As shown in Fig. 1, the encoder $f$ and decoder $g$ follow the structure of MAE (He et al., 2022), and $g \circ f$ is defined as the purifier in MAEP. Nevertheless, our MAEP makes certain modifications to meet its distinct purpose in that instead of solely focusing on learning a representation, the purifier is specifically designed to effectively remove perturbations from adversarial images.

**Patch Size $ps$.** In our experiment, as shown in Table 15, the patch size $ps$ (or patch area $ps \times ps$) of MAEP is a very important parameter in that it causes a large gap between different settings. Different from $ps = 14$ or $ps = 16$ in ViT (Dosovitskiy et al., 2020) and MAE (He et al., 2022), the patch size $ps = 2$ for MAEP should be small enough to get a considerable performance. The main reason is that MAE needs a sufficient hard task to learn patch representations and the performance of class prediction can be increased by finetuning the model for downstream tasks, but, different from MAE, our MAEP needs to reconstruct an image with only the masked region.

**Masking Ratio $r$.** Unlike the approaches of requiring masking 75% of image patches in MAE (He et al., 2022) and 15% of words in BERT (Kenton & Toutanova, 2019), it suffices for MAEP to empirically employ a 50% patch masking strategy to enhance performance. We opt to reduce the mask ratio $r$ because reconstructing a 75% masked image is too challenging. On the contrary, increasing the mask ratio beyond 15% can potentially augment model performance in auxiliary ways.

Further elaboration on these settings is shown in Table 15. We can see that MAEP degenerates into DISCO with the purifier loss in Eq. (2) when masking ratio $r = 0$. The main difference between DISCO and MAEP is the model structure and the performance difference can be seen from the column with masking ratio $r = 0$. To be specific, when $ps = 2$ and $r = 0$, the average accuracy of MAEP surpasses DISCO due to the MAE structure. Under $ps = 2$, as the masking ratio $r$ increases, the average accuracy of MAEP increases until $r = 0.5$ due to the masking mechanism. When $r = 0.75$, the average accuracy of MAEP falls because of the hardness of image reconstruction.

| Method | Patch size ($ps$) | $r=0$ | | $r=0.25$ | | $r=0.50$ | | $r=0.75$ | |
|---|---|---|---|---|---|---|---|---|---|
| | | Clean | Robust | Clean | Robust | Clean | Robust | Clean | Robust |
| WRN28-10 | - | 94.78 | 0 | - | - | - | - | - | - |
| + DISCO (Ho & Vasconcelos, 2022) | - | 89.26 | 85.33 | - | - | - | - | - | - |
| + MAEP | 2 | 90.49 | 86.80 | 93.10 | 84.40 | 92.30 | 88.73 | 92.70 | 78.70 |
| | 4 | 90.95 | 84.90 | 90.83 | 85.10 | 92.10 | 80.20 | 92.80 | 62.55 |
| | 8 | 90.86 | 63.99 | 91.77 | 60.31 | 92.68 | 55.75 | 91.61 | 67.00 |

Table 15: Performance of MAEP with different patch sizes ($ps$) and masking ratios ($r$) on CIFAR10 dataset under AutoAttack with $\epsilon_\infty = 8/255$.

## 11  SIMILARITY BETWEEN THE PURIFIED IMAGE AND ORIGINAL IMAGE

Both the PSNR and SSIM (Wang et al., 2004) metrics were used to measure the similarity between the adversarial and purified adversarial images, denoted as "Adv_PSNR" and "Adv_SSIM," respectively. For benign (clean) images and their purified counterparts, they are denoted as "Clean_PSNR" and "Clean_SSIM," respectively. As shown in Tables 16 and 17, our method MAEP almost obtains better purified (clean/adversarial) image quality in terms of PSNR and SSIM under the condition of maintaining a higher clean and robust accuracy than the methods used for comparison. Fig. 3 shows the visual results.

## 12  MORE RESULTS WITH ATTACK BUDGET $\epsilon_2 = 0.5$

In this section, we provide more results with $\ell_2$ attack of $\epsilon_2 = 0.5$, in comparison with $\epsilon_\infty = 8/255$, in Table 18. The results of ScoreOpt-O attract our attention as ScoreOpt-N, an advanced version of ScoreOpt-O, was demonstrated to exhibit better performance in the original paper (Zhang et al., 2023). As can be seen from Table 16 to Table 18, ScoreOpt-N indeed achieves higher SSIM and PSNR scores but ultimately leads to overall lower accuracy. Based on these observations, we conjecture the

| Defense Methods | Clean_PSNR (↑) | Robust_PSNR (↑) | Avg. PSNR (↑) | Attacks |
|---|---|---|---|---|
| DISCO (Ho & Vasconcelos, 2022) | 33.5802 | 36.2641 | 34.922 | AutoAttack ($\epsilon_\infty = 8/255$) |
| ScoreOpt-N (Zhang et al., 2023) | 28.6164 | 29.0951 | 28.8557 | AutoAttack ($\epsilon_\infty = 8/255$) |
| ScoreOpt-O (Zhang et al., 2023) | 22.7722 | 23.1615 | 22.9668 | AutoAttack ($\epsilon_\infty = 8/255$) |
| MAEP | **34.2392** | **36.3695** | **35.3044** | AutoAttack ($\epsilon_\infty = 8/255$) |
| DISCO (Ho & Vasconcelos, 2022) | 33.5802 | **34.475** | 34.0278 | AutoAttack ($\epsilon_2 = 0.5$) |
| ScoreOpt-N (Zhang et al., 2023) | 28.6164 | 28.6639 | 28.6401 | AutoAttack ($\epsilon_2 = 0.5$) |
| ScoreOpt-O (Zhang et al., 2023) | 22.7722 | 22.9505 | 22.8614 | AutoAttack ($\epsilon_2 = 0.5$) |
| MAEP | **34.8307** | 34.2392 | **34.5349** | AutoAttack ($\epsilon_2 = 0.5$) |

Table 16: PSNR evaluation and comparison between our method and state-of-the-art methods. Classifier: WRN-28-10. Testing dataset: CIFAR-10.

| Defense Methods | Clean_SSIM (↑) | Robust_SSIM (↑) | Avg. SSIM (↑) | Attacks |
|---|---|---|---|---|
| DISCO (Ho & Vasconcelos, 2022) | 0.9682 | 0.9834 | 0.9758 | AutoAttack ($\epsilon_\infty = 8/255$) |
| ScoreOpt-N (Zhang et al., 2023) | 0.8998 | 0.9062 | 0.9030 | AutoAttack ($\epsilon_\infty = 8/255$) |
| ScoreOpt-O (Zhang et al., 2023) | 0.7442 | 0.7581 | 0.7512 | AutoAttack ($\epsilon_\infty = 8/255$) |
| MAEP | **0.9723** | **0.9841** | **0.9782** | AutoAttack ($\epsilon_\infty = 8/255$) |
| DISCO (Ho & Vasconcelos, 2022) | 0.9682 | **0.9743** | 0.9712 | AutoAttack ($\epsilon_2 = 0.5$) |
| ScoreOpt-N (Zhang et al., 2023) | 0.8998 | 0.8999 | 0.8999 | AutoAttack ($\epsilon_2 = 0.5$) |
| ScoreOpt-O (Zhang et al., 2023) | 0.7442 | 0.7534 | 0.7488 | AutoAttack ($\epsilon_2 = 0.5$) |
| MAEP | **0.9764** | 0.9723 | **0.9744** | AutoAttack ($\epsilon_2 = 0.5$) |

Table 17: SSIM evaluation and comparison between our method and state-of-the-art methods. Classifier: WRN-28-10. Testing dataset: CIFAR-10.

accuracy of ScoreOpt-O would be reduced if its resultant PSNR and SSIM are maintained to be as high as those obtained in ScoreOpt-N and MAEP.

| Model | Clean Acc. (%) | Robust Acc. (%) | Avg. Acc. (%) | Attacks |
|---|---|---|---|---|
| WRN-28-10 | 94.78 | 0 | 47.39 | AutoAttack ($\epsilon_\infty$ / $\epsilon_2$) |
| + DISCO (Ho & Vasconcelos, 2022) | 89.26 | 85.33 | 87.29 | AutoAttack ($\epsilon_\infty = 8/255$) |
| + DiffPure (Nie et al., 2022) | 88.15±2.70 | 87.29±2.45 | 87.72±2.57 | AutoAttack($\epsilon_\infty = 8/255$) |
| + ScoreOpt-N (Zhang et al., 2023) | 91.31 | 81.79 | 86.55 | AutoAttack($\epsilon_\infty = 8/255$) |
| + ScoreOpt-O (Zhang et al., 2023) | 89.18 | **89.01** | 89.09 | AutoAttack($\epsilon_\infty = 8/255$) |
| + MAEP | **92.30** | 88.73 | **90.52** | AutoAttack($\epsilon_\infty = 8/255$) |
| + DISCO (Ho & Vasconcelos, 2022) | 89.26 | 88.53 | 88.89 | AutoAttack ($\epsilon_2 = 0.5$) |
| + DiffPure* (Nie et al., 2022) | - | 90.37±0.24 | - | |
| + ScoreOpt-N (Zhang et al., 2023) | 89.98 | 89.11 | 89.54 | AutoAttack ($\epsilon_2 = 0.5$) |
| + ScoreOpt-O (Zhang et al., 2023) | **92.61** | **92.19** | **92.40** | AutoAttack ($\epsilon_2 = 0.5$) |
| + MAEP | 92.13 | 91.04 | 91.58 | AutoAttack ($\epsilon_2 = 0.5$) |

Table 18: Robustness evaluation and comparison between our method and state-of-the-art methods. sterisk (*) indicates that the results were excerpted from the papers. Classifier: WRN-28-10. Testing dataset: CIFAR-10.

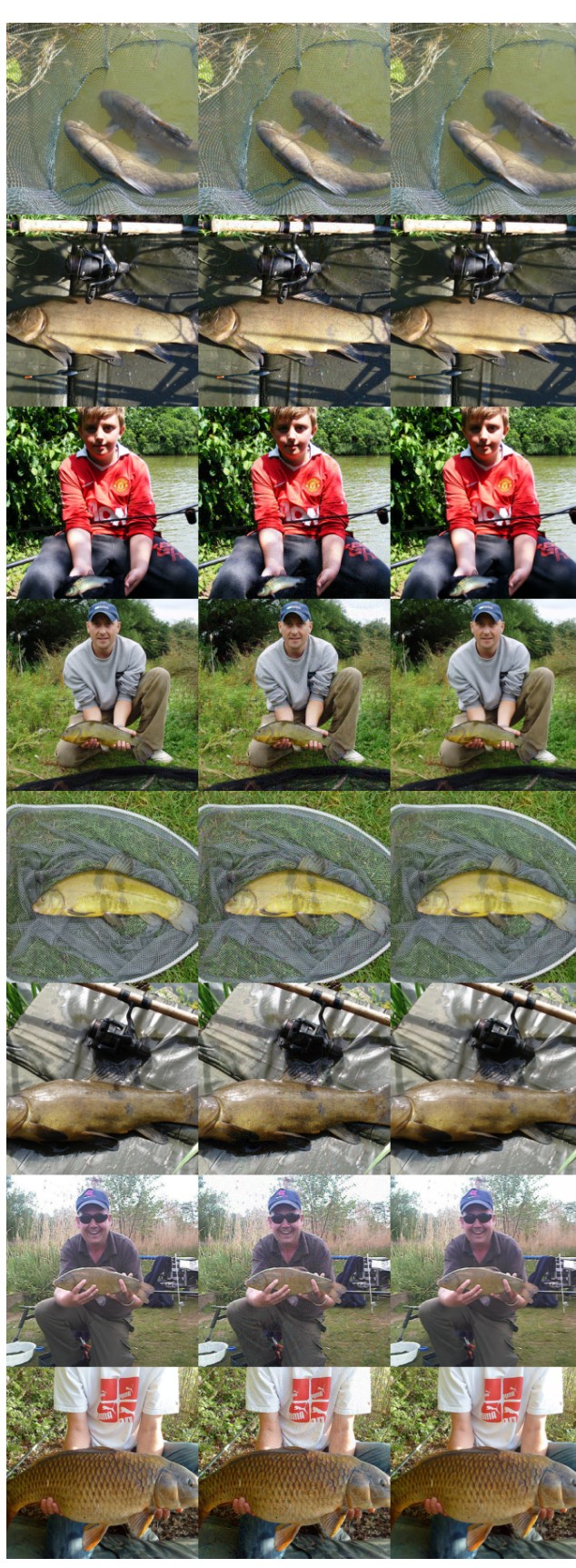

Figure 3: Comparison of clean images (left), adversarial images (middle), and purified images (right) by MAEP under AutoAttack. The MAEP is trained on CIFAR10 and directly tested on ImageNet without any finetuning.