# OpenReview forum: "Adversarial Masked Autoencoder Purifier with Defense Transferability"
_ICLR.cc/2025/Conference — ICLR 2025 Conference Withdrawn Submission_

### Official Review · Reviewer_wX9U · 2024-10-28

**Soundness:** 2
**Presentation:** 3
**Contribution:** 3
**Rating:** 3
**Confidence:** 4

**Summary:**

To better defend against adversarial attacks, the paper proposes a Masked AutoEncoder Purifier (MAEP), which utilizes the MAE model as a purifier for adversarial purification. MAEP achieves promising adversarial robustness, it particularly features model defense transferability and attack generalization without relying on using additional data that is different from the training dataset. Additionally, MAEP has a shorter inference time and can effectively mitigate the trade-off between robustness and accuracy.

**Strengths:**

1. The presentation of the paper is clear.
2. The paper conducts experiment across multiple datasets.
3. In the evaluation on ImageNet, the MAE-based purifier trained on CIFAR-10 even outperforms the diffusion-based purifier that is pre-trained on ImageNet.

**Weaknesses:**

1. After a simple search with the keywords “masked autoencoder adversarial purification” I found many similar papers [1,2,3], indicating that the idea of MAE-based purification was not first introduced in this paper. However, these papers are not discussed in the original manuscript, so I suggest that the author should conduct a more thorough investigation of this field.

2. The experimental results shown in the manuscript far exceed the current SOTA results, therefore I am concerned that some settings in the paper differ from standard settings.
a) Under the setting of using WRN-28-10 as a classifier on CIFAR-10 to defend against AutoAttack, the original DiffPure paper reports a robust accuracy of only 70.64%, while the current manuscript claims a robust accuracy as high as 87.29%.
b) Moreover, I notice that the results for most methods on PGD and AutoAttack (AA) are very close, as shown in Table 6. The average accuracy difference between defending against these two attacks for all methods is less than 1.5%, which is completely different from the results reported in [4,5]. In the l-inf norm threats, the robust accuracy difference between defending against PGD and AA is nearly 15%.
c) Due to the presence of the random factor, it would consider using PGD+EOT and AA (rand) for evaluation, instead of the PGD and AA (standard) used in the manuscript.
d) Removing perturbations by merely missing pixels will lead to low robust accuracy against unseen threats, which is mentioned by [6], and further discussed by [2]. To more effectively AP, [2] introduced Gaussian noise to ensure that the perturbation on each pixel is destroyed. However, MAEP maintains a high level of robust accuracy against unseen threats without the introduction of Gaussian noise, which is a bit unexpected.
e) Questions: How to generate the adversarial examples in the evaluation? What is the target of the attacks —is it the classifier model or the purifier model?

3. I notice that the MAEP model trained on CIFAR-10 is directly and efficiently applied to ImageNet and achieves performance exceeding that of the pre-trained diffusion model. I am curious about how this is achieved, specifically how a model trained on low-resolution images is applied to high-resolution images. One obvious question is how the two types of input images with different sizes are handled. Is image downsampling and upsampling used? Also, what is the specific pipeline for this process?

[1] Adversarial Purification of Information Masking. 2023.
[2] Denoising Masked AutoEncoders Help Robust Classification. 2023.
[3] MAEDefense: An Effective Masked AutoEncoder Defense against Adversarial Attacks. 2023.
[4] Robust Evaluation of Diffusion-Based Adversarial Purification. 2023.
[5] Robust Classification via a Single Diffusion Model. 2024.
[6] Trust-no-pixel: A remarkably simple defense against adversarial attacks based on massive inpainting. 2022.

**Questions:**

See Weaknesses.

---

### Official Review · Reviewer_kHeL · 2024-11-02

**Soundness:** 3
**Presentation:** 2
**Contribution:** 2
**Rating:** 5
**Confidence:** 4

**Summary:**

This work is the first to study an adversarial purifier based on Masked Autoencoders (MAE), proposing finetuning for test-time purification and discovering that MAEs exhibit good transferability in adversarial purification.

**Strengths:**

- Explores the effectiveness of using MAE for purification.
- Identifies transferability in the purification process, which is an interesting observation.
- Employs train-test discrepancy, which is an empirically effective strategy.

**Weaknesses:**

- I remain skeptical about the **necessity of using MAE**, given that purification methods based on diffusion models already perform well. Diffusion models, trained by adding Gaussian noise, align more closely with the nature of adversarial perturbations, whereas the training method (masked modeling) for MAE significantly differs from the adversarial perturbation approach. Generally, diffusion models exhibit better generative capabilities than MAE, leading me to question if MAE is a good choice. Could you provide the insight of leveraging MAE (instead of the diffusion model) for this specific task.
- The paper highlights improved transferability discovered by the authors but **lacks an explanation for this finding**. Could you provide like the learned representations, ablation studies to isolate which components of MAE contribute most to the improved transferability, or a theoretical analysis or hypothesis for this phenomenon.
- **Lack of ablation studies** like exploring the effects of applying finetuning strategies on diffusion models.
- **Minor**
  - The Methods section contains many experimental results. I believe this section should discuss why the proposed method exhibits transferability, rather than merely demonstrating it through results. The arrangement of content between the Methods and Experimental sections could be improved.
  - The PDF does not support hyperlinking citations and references.

**Questions:**

please see weaknesses

---

### Official Review · Reviewer_8mEr · 2024-11-02

**Soundness:** 1
**Presentation:** 2
**Contribution:** 1
**Rating:** 3
**Confidence:** 4

**Summary:**

This paper introduces the Masked AutoEncoder Purifier (MAEP), an innovative approach to adversarial defense that utilizes a Masked AutoEncoder (MAE) for purifying adversarial samples. Unlike diffusion-based adversarial purifiers, MAEP does not require additional training data and offers both defense transferability and attack generalization. By leveraging the MAE's masking mechanism, MAEP can effectively purify adversarial inputs while maintaining a high clean and robust accuracy balance. The method shows state-of-the-art performance on CIFAR10, even when tested on a higher-resolution dataset like ImageNet, surpassing existing diffusion-based models.

**Strengths:**

1. **Novelty in Using MAE**: MAEP represents an interesting attempt at using Masked AutoEncoder (MAE) as a purifier for adversarial samples in adversarial defense.

2. **Improved Accuracy Gap**: MAEP demonstrates strong performance in experiments, achieving a significant reduction in the gap between clean and robust accuracy.

3. **Theoretical Insights**: The paper provides concise mathematical reasoning, showcasing the authors' unique insights and streamlined analysis.

**Weaknesses:**

1. **Outdated Evaluation Techniques**: The evaluation methods used in this paper are outdated. Several recent papers [1,2,3] have demonstrated that the robustness of diffusion-based purification methods has been significantly overestimated due to insufficient evaluation techniques. All experiments in this paper are based on evaluations from nearly two years ago, leading to an overestimation of robustness for diffusion-based methods by more than 20%. Additionally, to properly assess the robustness of defenses, accurate gradient-based attack methods should be used; otherwise, the evaluations risk suffering from obfuscated gradients, as highlighted by Athalye et al. [4]. The high adversarial robustness values around 90% shown in Tables 2 and 5 appear overly optimistic.

   - **References**:
      - [1] Lee, Minjong, and Dongwoo Kim. "Robust evaluation of diffusion-based adversarial purification." *arXiv preprint arXiv:2303.09051* (2023).
      - [2] Kang, Mintong, Dawn Song, and Bo Li. "DiffAttack: Evasion Attacks Against Diffusion-Based Adversarial Purification." *Advances in Neural Information Processing Systems 36* (2023).
      - [3] Chen, Huanran, et al. "Robust Classification via a Single Diffusion Model." *ICML* (2024).
      - [4] Athalye, A., Carlini, N., Wagner, D. "Obfuscated gradients give a false sense of security: Circumventing defenses to adversarial examples." *International Conference on Machine Learning*, PMLR, 2018.

2. **Misleading Claims on Diffusion Model Efficiency**: The paper claims that “In contrast to most prior studies that rely on the diffusion model for test-time defense to remarkably increase the inference time,” suggesting that diffusion-based methods are inherently slow. However, diffusion models can achieve one-step denoising [5], and recent work has made significant advances in accelerating diffusion processes. For example, several recent studies [6,7,8] propose methods for efficient, training-free acceleration of diffusion models, challenging the paper’s claim that diffusion models are universally slow for test-time defense.

   - **References**:
      - [5] Carlini, N., Tramer, F., Dvijotham, K. D., et al. "(Certified!!) Adversarial robustness for free!" *arXiv preprint arXiv:2206.10550*, 2022.
      - [6] Xia, Mengfei, et al. "Towards More Accurate Diffusion Model Acceleration with a Timestep Aligner." *arXiv preprint arXiv:2310.09469* (2023).
      - [7] Li, Lijiang, et al. "AutoDiffusion: Training-Free Optimization of Time Steps and Architectures for Automated Diffusion Model Acceleration." *arXiv preprint arXiv:2309.10438* (2023).
      - [8] So, Junhyuk, Jungwon Lee, and Eunhyeok Park. "FRDiff: Feature Reuse for Universal Training-free Acceleration of Diffusion Models." *arXiv preprint arXiv:2312.03517* (2023).

**Questions:**

1. **Re-evaluation of DiffPure**: Please re-evaluate DiffPure using updated methods as outlined in the previously mentioned papers, to ensure accurate robustness assessment.

2. **Accurate Gradient-based Evaluation**: For evaluating your own defense method, please employ accurate gradient-based attacks to avoid issues related to obfuscated gradients.

3. **Clarification on Novelty**: Could you clarify the main differences between your work and the following two papers?
   - (1) Tsai, Y. Y., Chao, J. C., Wen, A., et al. "Test-time Detection and Repair of Adversarial Samples via Masked Autoencoder." *arXiv preprint arXiv:2303.12848*, 2023.
   - (2) Lyu, W., Wu, M., Yin, Z., et al. "MAEDefense: An Effective Masked AutoEncoder Defense against Adversarial Attacks." *2023 Asia Pacific Signal and Information Processing Association Annual Summit and Conference (APSIPA ASC)*, IEEE, 2023, pp. 1915-1922.

---

### Official Review · Reviewer_EwTL · 2024-11-04

**Soundness:** 2
**Presentation:** 2
**Contribution:** 2
**Rating:** 3
**Confidence:** 4

**Summary:**

In this paper, the authors explore the use of an adversarial purification framework based on the MAE architecture. They present an analysis of various performance metrics, including clean and robust accuracy, running time, and comparative evaluations against several state-of-the-art methods. Additionally, they provide experimental results across different datasets while proposing a new training loss function for better adversarial performance.

**Strengths:**

1. Innovative Use of the MAE Framework: The application of the MAE framework for adversarial purification demonstrates an approach that potentially leverages the strengths of this architecture to enhance model robustness.

2. Comprehensive Evaluation: The authors conduct extensive experiments, comparing the proposed method against several baselines, which enriches the understanding of its performance.

**Weaknesses:**

1. Table I Clarity: The meanings of the symbols in the last three columns of Table I are not defined in the caption. Providing clear explanations would improve the comprehensibility of the results.

2. Advantages of MAE Framework: The paper lacks a detailed discussion on the specific advantages of using the MAE framework for adversarial purification. Clarifying this would enhance the reader's understanding of its effectiveness.

3. Confusion Regarding Table IV Results: The results presented in Table IV are confusing. A clearer relationship should be established between these results and the design of the training loss function to aid comprehension.

4. Presentation of Accuracy Metrics: It is recommended to show both clean and robust accuracy simultaneously using a scatter diagram, which would provide a clearer visual representation of the performance trade-offs.

5. Inconsistencies in DiffPure Results: In Table 6, only the results of DiffPure are reported with standard deviation. It would be beneficial to clarify why other methods are not included. Additionally, based on the reported accuracies, DiffPure seems to outperform MAEP in some settings, yet this is not adequately discussed.

6. Limited Comparison Between MAEP Variants: The differences between MAEP and MAEP with LoRA are minimal, particularly noted in Table 7. A more thorough analysis of their comparative advantages would be useful.

7. Source Consistency for DiffPure Results: The results for DiffPure across different tables appear inconsistent, with some data excerpted from previous works while others are reproduced. Ensuring results come from a uniform source would facilitate a more standardized comparison.

8. Additional Testing Settings: The paper should report the results of DiffPure in Table 8 under the setting of training on CIFAR10 and testing on ImageNet to provide a comprehensive evaluation.

9. Running Time Analysis: While MAEP performs better than some baselines like DiffPure, it is less efficient than DISCO. A deeper discussion of running time implications in practical applications would be beneficial.

**Questions:**

1. Could you clarify the meanings of the symbols in the last three columns of Table I? Including this in the caption would significantly aid in understanding the results.

2. What are the specific advantages of utilizing the MAE framework for adversarial purification?

3. Can you provide a clearer explanation of the results presented in Table IV and how they relate to the training loss design?

4. Why does Table 6 only report DiffPure results with standard deviation? Could you provide a rationale for this approach?

5. Could you report DiffPure’s results from Table 8 under the setting of training on CIFAR10 and testing on ImageNet?

6. What accounts for the minimal performance differences observed between MAEP and MAEP with LoRA in Table 7?

---

### Note · Authors · 2024-11-15

I have read and agree with the venue's withdrawal policy on behalf of myself and my co-authors.